# Dual-energy three-compartment breast imaging for compositional biomarkers to improve detection of malignant lesions

Lambert T. Leong [1,2], Serghei Malkov[3], Karen Drukker[4], Bethany L. Niell[5], Peter Sadowski [6], Thomas Wolfgruber [1], Heather I. Greenwood[7], Bonnie N. Joe[7], Karla Kerlikowske[3,8], Maryellen L. Giger [4] & John A. Shepherd [1✉]

## Abstract

**Background** While breast imaging such as full-field digital mammography and digital breast tomosynthesis have helped to reduced breast cancer mortality, issues with low specificity exist resulting in unnecessary biopsies. The fundamental information used in diagnostic decisions are primarily based in lesion morphology. We explore a dual-energy compositional breast imaging technique known as three-compartment breast (3CB) to show how the addition of compositional information improves malignancy detection.

**Methods** Women who presented with Breast Imaging-Reporting and Data System (BI-RADS) diagnostic categories 4 or 5 and who were scheduled for breast biopsies were consecutively recruited for both standard mammography and 3CB imaging. Computer-aided detection (CAD) software was used to assign a morphology-based prediction of malignancy for all biopsied lesions. Compositional signatures for all lesions were calculated using 3CB imaging and a neural network evaluated CAD predictions with composition to predict a new probability of malignancy. CAD and neural network predictions were compared to the biopsy pathology.

**Results** The addition of 3CB compositional information to CAD improves malignancy predictions resulting in an area under the receiver operating characteristic curve (AUC) of 0.81 (confidence interval (CI) of 0.74–0.88) on a held-out test set, while CAD software alone achieves an AUC of 0.69 (CI 0.60–0.78). We also identify that invasive breast cancers have a unique compositional signature characterized by reduced lipid content and increased water and protein content when compared to surrounding tissues.

**Conclusion** Clinically, 3CB may potentially provide increased accuracy in predicting malignancy and a feasible avenue to explore compositional breast imaging biomarkers.

## Plain language summary

Breast cancers are detected by mammography. This study explored the use of a particular kind of mammography technique to obtain information about the composition of cancerous and non-cancerous breast tissue. This technique provided measures of lipid (fat), water, and protein content in addition to shape characteristics provided from standard mammography. Adding information about the composition of the tissue to its shape characteristics resulted in an increased ability to distinguish invasive cancerous tissue from unaffected surroundings. Invasive breast cancer tissues were also found to exhibit lower lipid, higher protein and higher water content when compared to other non-invasive, non-cancerous breast tissues in which cancer was suspected. Our findings highlight the added value of including the composition of breast tissue when deciding if biopsy of the suspicious tissue is warranted.

[1] Department of Epidemiology and Population Sciences, University of Hawaii Cancer Center, Honolulu, HI, USA. [2] Department Molecular Bioscience and Bioengineering, University of Hawaii at Manoa, Honolulu, HI, USA. [3] Departments Epidemiology and Biostatistics, University of California, San Francisco, San Francisco, CA, USA. [4] Department of Radiology, University of Chicago, Chicago, IL, USA. [5] Department of Diagnostic Imaging and Interventional Radiology, H. Lee Moffitt Cancer Center and Research Institute, Tampa, FL, USA. [6] Department of Information and Computer Science, University of Hawaii at Manoa, Honolulu, HI, USA. [7] Department of Radiology and Biomedical Imaging, University of California, San Francisco, San Francisco, CA, USA. [8] Department of Medicine, University of California, San Francisco, San Francisco, CA, USA. ✉email: johnshep@hawaii.edu

Breast cancer is the leading cause of cancer death among women globally[1]. Early detection with screening mammography has a beneficial impact on survival and has been shown to reduce cancer mortality[2–6]. However, the accuracy resulting from breast imaging technologies still has room for improvement. For instance, in the United States, 71% of biopsies do not result in a breast cancer diagnosis suggesting a modest specificity[7,8]. Furthermore, breast density affects the accuracy of full-field digital mammography (FFDM) since dense tissue can mask tumors, diminishing the sensitivity of mammography by 10–20% compared to women with fatty breasts[9]. Compared to FFDM, digital breast tomosynthesis (DBT) increases cancer detection rates and decreases recall rates. However, the added benefit of DBT is difficult to quantify and studies have demonstrated that, positive biopsy rates following screening DBT are similar to those following screening FFDM[9,10]. Also, in a registry study including over 1.5M screening mammograms from 46 registry sites, it was shown that women with the extremely dense breast tissue had neither reduced recall nor increased cancer detection rates for DBT compared to FFDM[11]. Improvements to sensitivity and specificity are needed and could result in an increase in detecting malignancies and reduction of unnecessary, benign biopsies.

The fundamental information that a radiologist uses, the attenuation of X-rays from a single exposure, has remained the same since the inception of breast imaging in 1913 (ref. [12]). Without additional information, mammography provides only relative radiopacity (i.e. tissue density relative to a background of fat) and lesion type, such as mass, asymmetry, distortion, or calcifications. Lesion classification is limited to detection of calcifications, which are often benign, as well as the shape and symmetry of high-density breast masses. Thus, lesion classification has limited reliably in predicting an invasive breast cancer. Computer-aided detection (CAD) software attempts to improve the diagnostic accuracy of mammography through the utilization of computer vision and artificial intelligence algorithms to automatically identify anomalies[13]. Yet, the fundamental information used by CAD is identical to the information radiologists use. While CAD has been shown to be clinically beneficial by some[14,15], others have shown that the addition of CAD had no significant improvement to screening sensitivity and specificity[16]. It is likely that the limit of diagnostically relevant information from radiologists and/or CAD has been reached with X-ray based, single-energy mammography, especially in women with dense breasts.

Additional diagnostic information can be obtained via contrast-enhanced mammography (CEM). Contrast imaging has demonstrated increased sensitivity to detect cancer due to differential vascularization of cancerous and benign tissue[17]. Invasive breast cancer typically presents as a mass of epithelial cells with a high degree of vascularization. IDC, and often DCIS due to its own vascularization, enhance on contrast imaging methods. However, these techniques still have low specificity because benign lesions also enhance with contrast[18–20]. Like mammography, the diagnostic information gained with contrast imaging is still based in the lesion morphology and structure of surrounding tissue. Since intravenous contrast can cause adverse effects, CEM is often used as a secondary imaging tool. Therefore, radiologists are often not afforded this information on the initial screening exam.

Radiomic features and imaging biomarkers based on tissue composition have the potential to address accuracy issues seen with current imaging techniques and technologies. Evidence suggests that the biology and atomic composition of malignant lesions differ from benign lesions and these differences manifest into macroscopically unique tissue compositions which are measurable with multispectral X-ray imaging[21,22]. First, invasive cancer is highly angiogenic and malignant tumors have been shown to consume lipids to sustain high rates of proliferation[23,24]. The central to peripheral microvasculature of the lesion differs significantly between normal tissue, fibroadenomas (FA) and different grades of invasive ductal carcinoma (IDC)[25,26]. Second, adipocytes, available at the tumor stromal interface, have demonstrated a pro-tumorigenic role for breast cancer[27]. Triple-negative cancers utilize and require fatty-acid oxidation leeched from the surrounding tissues. This has been observed using multispectral mammograms as a decrease in fat composition surrounding triple-negative cancers versus receptor-positive tumors[28]. Third, Cerussi et al.[29] found a 20% reduction in lipid, and 50% increase in water, content in invasive breast cancer versus normal breast tissue. A strong positive correlation ($R = 0.98$) between the macroscopic water concentration and the Scarff Bloom-Richardson Score (a histological grading scale ranging from 3 to 9 that accounts for tubule formation, nuclear pleomorphism, and mitosis counts) was also reported[30]. Fourth, invasive cancers have significantly lower X-ray attenuation than FAs that also lead to biopsy, suggesting a distinctly different composition between cancerous and benign masses[17].

The purpose of this study is to demonstrate that compositional profiles of the breast combined with CAD predictions can improve specificity of breast cancer detection. A dual-energy mammography technique known as 3-compartment breast (3CB) imaging was used to obtain the lipid–water–protein (LWP) fractions of the breast on a pixel-by-pixel basis. The 3CB scientific principals and imaging protocols have been previously presented[21,31] as well as the characteristics of malignant versus benign lesions[22,32]. To quantify the added clinical value of 3CB imaging, we compared the performance of CAD-based models to identify malignancies without and with 3CB lesion characterization. Malignant and non-malignant masses and hormone receptor status were further studied to better understand the biological mechanism which led to increased specificity of models that include 3CB composition. Compositional information from 3CB improved accuracy of malignancy predictions when compared to CAD and confirmed that invasive breast lesions have unique compositional signatures when compared to other lesion types.

## Methods

**Data acquisition.** The participants in this study were women identified from screening and diagnostic mammography populations at the University of California San Francisco (San Francisco, CA) and H. Lee Moffitt Cancer Center and Research Institute (Tampa, FL). Women who presented with Breast Imaging-Reporting and Data System (BI-RADS) diagnostic categories 4 or 5 and who were scheduled for breast biopsies were consecutively recruited for 3CB imaging. Demographics and characteristics of this studies population are detailed in Table 1. The 3CB imaging clinical study was approved by the institutional review board at all participating research sites (University of California, San Francisco, University of Chicago, and H. Lee Moffitt Cancer Center and Research Institute) and followed Health Insurance Portability and Accountability Act-compliant protocols. All study participants provided written informed consent.

In addition to clinical diagnostic mammograms, participants underwent further research imaging using the 3CB protocol prior to breast biopsy. FFDMs, 2D images, were acquired on Hologic Selenia systems (Hologic, Inc., Bedford, MA). In brief, the 3CB imaging protocol consisted of two images in succession: a clinical mammogram (autocontrast, autocompression release off) and a high-energy (HE) image acquired at 39 kVp (40 mAs, 3-mm

additional aluminum filtration). A calibration phantom was placed on top of the breast compression paddle during imaging to accurately estimate paddles compression depth, warp, and tilt from which exact submillimeter point thicknesses of the breast could be calculated[33]. With these three pieces of information (HE attenuation, low-energy (LE) attenuation, and local breast thickness) a system of three equations was solved which resulted in the LWP thicknesses at each pixel. Absolute accuracy of this technique has been previously verified using reference standards[21,34].

Pathology results were reported on all biopsies and radiologist delineated regions of interest (ROIs) for the mammographic abnormalities on presentation mammogram images. Participants were excluded if biopsy site annotation coordinates could not be correctly registered on presentation or 3CB images, if the lesion pathology was incomplete, or if the 3CB data set was incomplete. The 3CB protocol requires that images be acquired on calibration phantoms prior to patient imaging and the absence of calibration images or poor image quality, due to excessive movement between HE and LE image acquisition, resulted in an incomplete 3CB data set and exclusion. See Fig. 1. for a flowchart of study participant enrollment and derivation of final data set.

Table 2 stratifies ROIs by BI-RADS density categories.

**3CB feature extraction**. The 3CB LWP thickness maps were generated for all FFDM images and were used to quantify the

### Table 1 Participant stratification by age, BMI, BI-RADS density, and duration of hormone therapy.

|  | N | Percentage |
|---|---|---|
| Participants | 349 | 100 |
| **Age** | | |
| <40 | 20 | 6 |
| 40 to <50 | 120 | 34 |
| 50 to <60 | 118 | 34 |
| 60 to <70 | 57 | 16 |
| 70 to <80 | 30 | 9 |
| ≥80 | 4 | 1 |
| **BMI** | | |
| <18.5 | 9 | 3 |
| 18.5 to <25 | 120 | 34 |
| 25 to <30 | 92 | 26 |
| ≥30 | 128 | 37 |
| **BI-RADS** | | |
| A | 23 | 7 |
| **Density** | | |
| B | 130 | 37 |
| C | 162 | 46 |
| D | 34 | 10 |
| **Hormone therapy** | | |
| None | 321 | 92 |
| <5 years | 10 | 3 |
| ≥5 years | 18 | 5 |

### Table 2 Saparation of all 689 radiologist delineated ROIs by pathology and BI-RADS density.

| BI-RADS density | A | B | C | D | Total findings |
|---|---|---|---|---|---|
| Invasive | 21 | 33 | 45 | 4 | 103 |
| DCIS | 2 | 27 | 22 | 10 | 61 |
| Fibroadenoma | 8 | 49 | 41 | 18 | 116 |
| Other benign | 34 | 164 | 178 | 33 | 409 |
| **Total** | **65** | **273** | **286** | **65** | **689** |

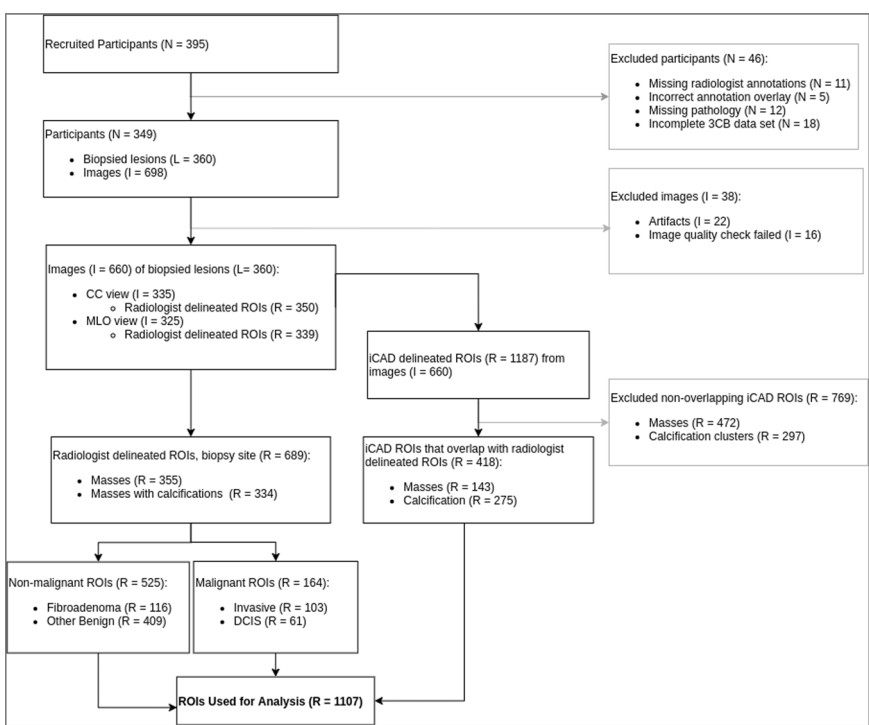

**Fig. 1 Overview of participants and data used for modeling and analysis.** Flow diagram detailing inclusion and exclusion of data used in the final analysis. This study includes 349 patients (*N*) which equates to 360 biopsy sites (L) and 660 mammographic images (I) which includes craniocaudal (CC) and mediolateral oblique (MLO) views. The 660 images contained 689 radiologist delineated region of interests (ROIs) (R) and 413 computer-aided detection (CAD) delineated ROIs agreed with radiologist delineated ROIs. The final data set contained all radiologist ROIs and agreeing CAD ROIs which results in 1107 ROIs.

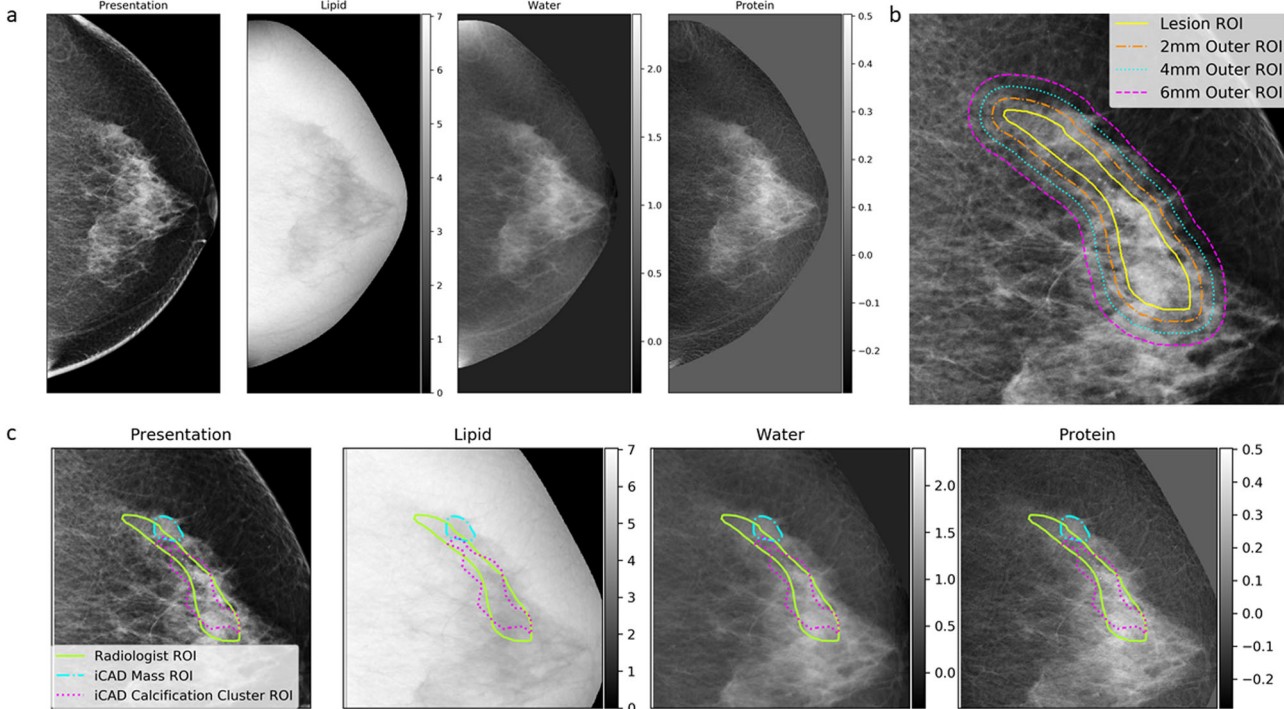

**Fig. 2 3CB, lipid, water, and protein, data, and regions of feature extraction. a** Full presentation mammogram image and the derived three-compartment breast (3CB) thickness maps. From left to right is the standard presentation craniocaudal mammogram used for reading by a radiologist, lipid thickness map, water thickness map, and protein thickness map. Grayscale colorbars, adjacent to 3CB thickness maps, indicate thickness in cm. **b** The composition of the background or tissue surrounding a lesion was measured progressively by capturing three outer regions extending from the border of the lesion (yellow solid line). The outer regions extend from the lesion border at distances of 2 mm (orange dot-dashed line), 4 mm (cyan dotted line), and 6 mm (magenta dashed line). **c** Computer-aided detection (CAD) delineations that had some agreeance with radiologist region of interest (ROIs) (yellow line) were included in the final data set. CAD delineates suspicious masses (cyan dot-dashed line) and calcification clusters (magenta dotted line). Outer regions for all ROIs (radiologist and CAD delineated) were calculated but not displayed in this sub-figure for easy viewing.

composition within the radiologist delineated ROIs. Standard presentation images and their fully registered 3CB compositional maps can be observed in Fig. 2a. Note that the 3CB images are thickness maps where each pixel corresponds to a thickness, in centimeters, of a given composition. Recall, we are investigating the diagnostic impact of independently adding compositional information to morphological features already existing in standard clinical FFDM. To abstract compositional information away from morphological features, we computationally extracted nine measurements to quantify the composition within a given region. These nine measurements included the mean, median, standard deviation, minimum, maximum, kurtosis, skew, total, and percentage value of all pixels contained within a ROI.

Three additional outer ROIs were derived from the lesion ROI to capture the background or tissue immediately surrounding a lesion, see Fig. 2b. Each outer region captured all pixels extending 2 mm from the border of the previous region. Therefore, the first, second, and third outer regions contain all pixels extending from the edge of the lesion ROI out to 2 mm, the edge of the first outer region out to 2 mm, and the edge of the second outer region out to 2 mm, respectively. In other words, in relation to the lesion border, the first, second, and third outer regions measure 0–2 mm, 2–4 mm, and 4–6 mm, respectively. For each lesion, we obtained nine compositional measurements from four ROIs (lesion and three outer regions) on each of the three compositional images (3CB LWP maps) which resulted in 108 compositional features per lesion ROI.

**Clinical CAD lesion detection.** Low-energy, standard FFDMs were processed using commercial CAD software (SecondLook,

version 7.2, iCAD, Nashua, NH) to identify suspicious masses and calcifications. The CAD software utilizes a proprietary algorithm to delineate suspicious ROIs for masses and individual calcifications as well as assigns a probability of malignancy for each delineation. Note that for input to our analysis, we used the calcification cluster ROI rather than each individual calcification ROI. Calcification cluster ROIs were calculated using the convex hull or minimum envelope which encompasses all calcifications associated with a cluster. Therefore, CAD delineated ROIs, used in our final analysis, may consist of either a suspicious mass or a calcification cluster.

**Predictive modeling with morphology and 3CB.** The final data set, consisting of compositional features extracted from ROIs, was split by patient ID into a train, validation, and test set using a 60, 20, and 20% split. The data were split by patient ID such that all ROIs for a given patient remained exclusively in one of the three datasets. These data split condition ensured no data leakage and ROIs from a single patient, which are highly correlated, did not end up in both the training and test set, for example. To reiterate, the train, validations, and test datasets contained their own unique subset of patients and patient ROIs and the test set contained 20% of the patients.

A neural network model was trained to predict malignancy probability from the 108 extracted 3CB features and the prediction from CAD. CAD predicts probabilities of malignancy rather than specific lesion type. To compare against CAD performance, target labels were created for our data set which combined BN and FA pathologies into a non-malignant label. ROIs with DCIS and IDC pathologies were also combined into a

new malignant label. The final model was trained to output these new targets or probability of malignancy. Additional details on the neural network architecture, tuning, and hyperparameters optimization can be found in the extended Methods Section and Supplementary Fig. 1.

**Quantifying the added diagnostic benefit of 3CB for malignancy prediction**. The benefit of 3CB composition was evaluated using the following metrics area under the receiver operating characteristic (ROC) curve (AUC), the integrated discrimination improvement (IDI) and the net reclassification improvement (NRI)[35,36]. All metrics were computed on the unsee, independent hold out test set. The 95% confidence intervals (CI) and the mean AUC were computed via 1000 rounds of bootstrapping. All samples were selected randomly for each bootstrap round and thus the number of replacements were also random for each bootstrap round.

IDI and NRI offer additional insight into the benefits of new biomarkers beyond AUC comparison. Performance differences between a reference model and a new model, which contains the added biomarkers are evaluated across all calculated risks. The NRI measures the number of cases correctly reclassified by the new model while the IDI also takes into account the magnitude of the change in discrimination slopes. The NRI is the sum of the events NRI and the non-events NRI. In the context of this study, events and non-events correspond to malignancies and benigns, respectively. Therefore, the NRI captures the percent improvement of correctly classified malignancies and benigns by the new model which includes 3CB features. The IDI is the sum of the integrated sensitivity (IS) and the integrated 1-specificity (IP). The IS is the difference in the mean probability of malignancy for those with cancer between CAD and the neural network while the IP is the difference in the mean probability of malignancy for those with benign masses between CAD and the CAD + 3CB neural network models. The NRI and IDI changes were evaluated with respect to BI-RADS assessment categories as to investigate the clinical implications of 3CBs improvement. The BI-RADS categories of interest were 3, 4a, 4b, and 4c, with risk threshold of 2%, 10%, 50%, and 95%[37], respectively.

**Lesion composition characterization**. Using quantitative methods, we further investigate compositional differences among the four different lesion pathologies. To quantify these unique signatures, the median LWP values from each of the surrounding outer region ROIs were subtracted from the median LWP values from within the lesion ROI. Only radiologist drawn ROIs delineating biopsy sites were included in this analysis. Microcalcifications are present in many of the mammograms and although they are not composed of lipid, water, or protein, they can produce a high water and protein signal in the 3CB thickness maps. Therefore, the median pixel values were used to mitigate the influence microcalcifications have on the mean composition within an ROI. Lesion signatures were stratified by pathology and compositional component type (i.e. lipid, water, or protein).

Our model predicts probability of malignancy rather than lesion type, so malignant and non-malignant types were grouped for this analysis. The average signature for malignant lesion types (DCIS and IDC) and the average signature between non-malignant types (BN and FA) were computed for all outer ROIs. Differences between the malignant and non-malignant compositions were computed and $p$ values were derived using Welch's test for unequal variance.

We also looked at possible correlations between invasive cancers and patient hormone receptor status. It is hypothesized that cancers of different receptor type have unique compositional signatures due to utilization of exogenous fatty acids for sustained growth[28,38,39]. To investigate, we compared the composition of IDC lesions to their background and stratified each lesion by hormone receptor status. We compared triple-negative receptor lesions to all receptor-positive lesions: estrogen receptor, progesterone receptor, human epidermal growth factor receptor 2, or any combination of the three. Differences between the triple-negative and receptor-positive lesions composition were also computed, and $p$ values were derived using Welch's test for unequal variance.

**Reporting summary**. Further information on research design is available in the Nature Research Reporting Summary linked to this article.

## Results

**Final participant data**. This work reports on results from 349 participants after exclusions. Participants were imaged in both the craniocaudal (CC) and mediolateral oblique (MLO) views which resulted in 698 images of 349 women participants. After image exclusion, 660 images remained which contained a total of 689 radiologist delineated ROI of biopsy sites for which pathology was reported. The 689 ROIs consisted of 103 IDC, 61 ductal carcinoma in situ (DCIS), 116 FA, and 409 other benign (BN), see Fig. 1a. In addition, CAD delineated 1187 ROIs from the 660 images. Only 418 CAD delineated ROIs had a 25% or greater overlap with the radiologist delineated biopsy sites. The 769 non-overlapping CAD ROIs were excluded from our analysis because they did not overlap biopsy sites, and thus pathology diagnosis could not be confirmed. Of the patients with DCIS pathologies, CAD failed to delineate any ROI on one patient resulting in a complete miss. CAD missed one delineation for the CC view for one patient and missed another delineation on the MLO view for another patient. In total, four DCIS ROIs were not identified by CAD. Of patients with IDC pathologies, CAD completely missed delineations on seven patients, missed three delineations on the MLO view, and one delineation for the CC view. In total, 18 IDC lesions were not identified by CAD but were delineated by the radiologists. These lesions were assigned a CAD probability of malignancy of zero and are present in the final training data set only. The final data set consisted of 108 3CB features and a CAD probability of malignancy on 1107 ROIs (689 radiologist and 418 CAD delineated, see Fig. 1a) from 349 patients.

**Model performance with morphology and morphology plus 3CB**. On the unseen, independent hold out test set, the commercial CAD output of probability of malignancy resulted in a mean AUC curve of 0.69 and a CI of 0.60–0.78. On this same test set, the neural network model, which utilized both morphological features captured by CAD and compositional features derived from 3CB, resulted in a mean area under the curve (AUC) of 0.81 and CI of 0.74–0.87 (see Fig. 3a).

We plot the IDI curves in Fig. 3b and the CAD (dashed lines) and CAD + 3CB (solid lines) represent the reference and new model, respectively. The space or black shaded area between the black dashed and solid lines indicates the IS which is −1.06%. The space or red shaded area between the black dashed and solid lines indicates the IP which is 13.17.

Using the predicted malignancy probabilities output from the CAD and CAD + 3CB, we investigate the NRI with respect to BI-RADS assessment categories. The vertical dash lines in Fig. 3b indicate the border between different BI-RADS categories. The NRI for malignant lesions was 2, 4, 16, and −45% at the BI-RADS 3/4a, 4a/4b, 4b/4c, 4c/5 borders. The NRI for benign lesion at the aforementioned BI-RADS borders was 2, 12, 13, and 17% resulting in a total NRI at each border of 4, 15, 29, and −28%.

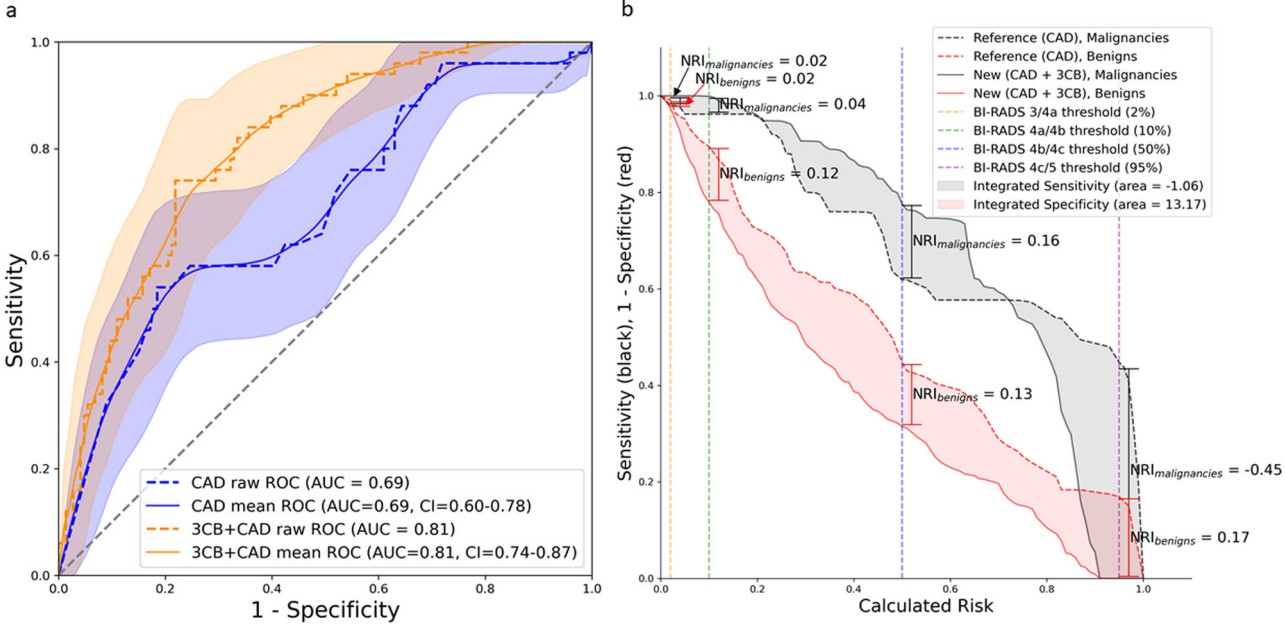

**Fig. 3 Improved performance on unseen test set when adding 3CB compositional information. a** Adding three-compartment breast (3CB) features to computer-aided detection (CAD) (orange) results in an area under the receiver operating characteristic (ROC) curve (AUC) of 0.81, standard deviation (SD) of 0.03, when compared to CAD alone (blue), AUC of 0.69, SD of 0.04. Mean curves (solid lines) and 95% confidence intervals (shaded regions) were computed via 1000 bootstrap samples. **b** The integrated sensitivity (IS), black shaded region between solid and dashed lines, indicates the change in sensitivity with the addition of 3CB. The integrated 1-specificity (IP), red shaded region between solid and dashed lines, indicates the change in specificity with the addition of 3CB. The integrated discrimination improvement (IDI) is the sum of the IS and IP ($-1.06 + 13.17$) which is 12.11 and a positive IDI indicates that predictive models benefit from the addition of 3CB. The borders of the Breast Imaging-Reporting and Data System (BI-RADS) assessment categories are indicated by the vertical dashed lines. Net reclassification improvement (NRI) for events or cancers (black) and non-events or benigns (red) are calculated at each BI-RADS border to demonstrate 3CBs effect on specificity with respect to each BI-RADS category.

**Table 3 Net reclassification with respects to BI-RADS risk categories.**

| Reference (CAD) | Events (CAD + 3CB) | | | | | Non-events (CAD + 3CB) | | | | |
|---|---|---|---|---|---|---|---|---|---|---|
| BI-RADS thresholds (risk range) | 3 (0-≤2%) | 4a (2-≤10%) | 4b (10-≤50%) | 4c & 5 (>50%) | Total | 3 (0-≤2%) | 4a (2-≤10%) | 4b (10-≤50%) | 4c & 5 (≥50%) | Total |
| 3 (0-≤2%) | 0 | 0 | 0 | 1 | 1 | 0 | 0 | 0 | 0 | 0 |
| 4a (2-≤10%) | 0 | 0 | 1 | 0 | 1 | 2 | 2 | 8 | 3 | 13 |
| 4b (10-≤50%) | 0 | 0 | 9 | 8 | 17 | 0 | 6 | 51 | 9 | 66 |
| 4c & 5 (≥50%) | 0 | 0 | 11 | 20 | 31 | 0 | 5 | 47 | 13 | 65 |
| Total | 0 | 0 | 21 | 29 | 50 | 2 | 13 | 106 | 25 | 146 |

This table shows that adding 3CB allows for more accurate BI-RADS classification, as determined by probability of malignancy, for lesions with both malignant and non-malignant pathologies or events and non-events. The NRI for events and non-events is $-0.02$ and 0.25. The overall NRI, which is the sum of NRI events and non-events, is 0.25.

A breakdown of each test set ROIs classification by both reference CAD and new CAD + 3CB models as well as their NRIs are presented in Table 3.

**Lesion composition characterization.** Evidence demonstrates that malignant, particularly IDC, lesions have unique biological and compositional characteristics which may have contributed to better model performance. To investigate further, 3CB thickness heat maps for lesions of each type (BN, FA, DCIS, IDC) which resulted in high NRIs were generated for Fig. 4. Using the visible light color spectrum ordering convention, red indicates higher quantities of a given tissue component and quantities decrease as colors move towards violet. All lesion types, except DCIS, appear to have higher concentrations of protein and water relative to their background or surrounding tissue. Additionally, all lesions appear to have less lipid than surrounding parenchyma, and

invasive lesions contain considerably less lipid compared to their surroundings. The invasive lesions in particular appear to have a noticeably higher water signal.

Figure 5 shows that all lesions contain less lipid when compared to their background as indicated by negative median values in the box and whisker plots. The IDC lipid signature is strongest and distinctly different than the other lesion types. BN, FA, and DCIS lesions tend to have less water when compared to the surrounding tissue; however, IDC lesions show an increase in water content. All lesion types have a higher protein signature when compared to the three outer surrounding regions and IDC lesions show the largest signature. As demonstrated by the circumferential regions of interest, the protein content increases from the background reaching a peak in the lesion itself.

Differences between the malignant and non-malignant compositions are indicated by the space between the blue and orange

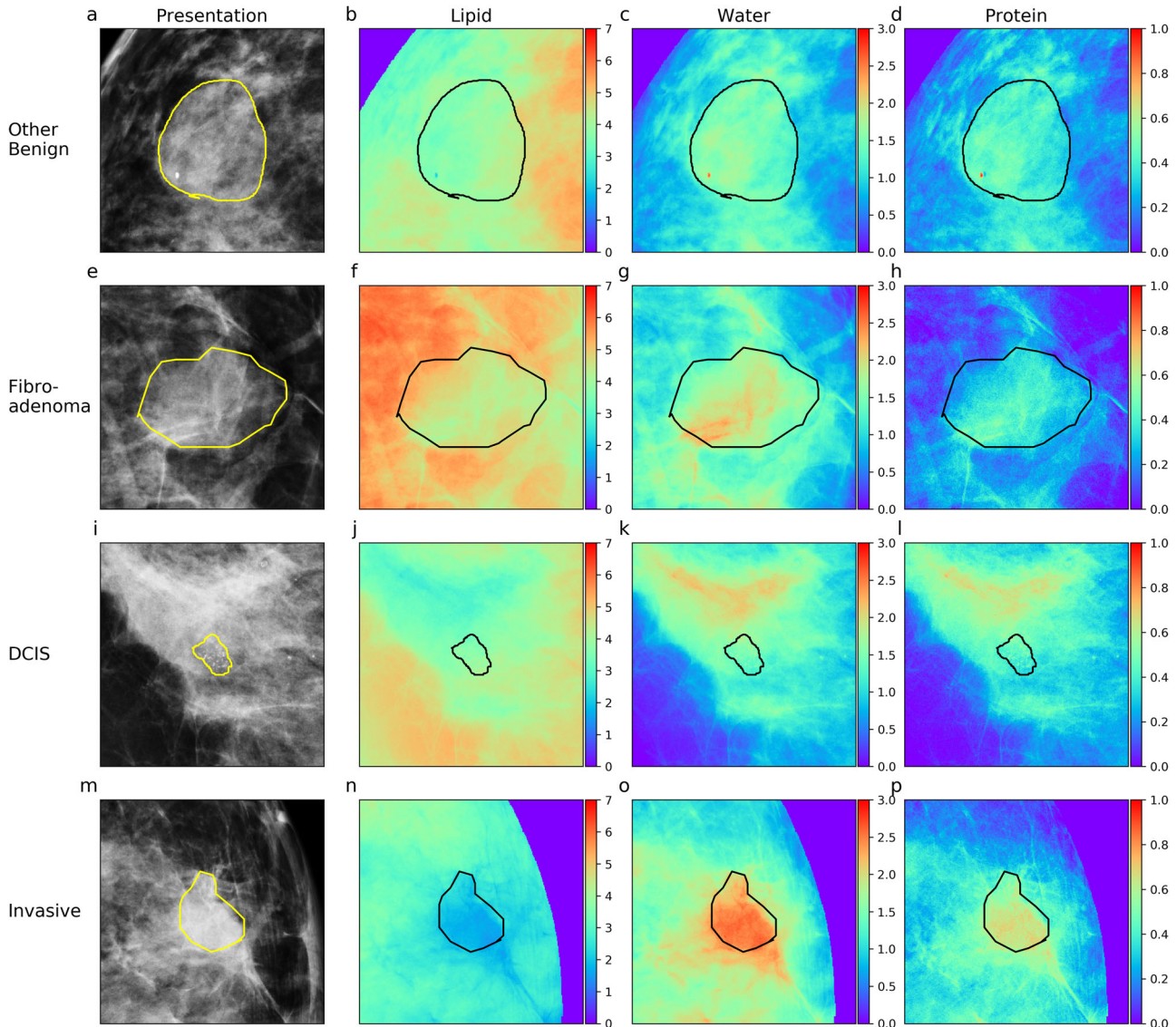

**Fig. 4 Compositional heat maps of all lesion pathologies.** Each row consists of a lesions with a different pathology. **a–d** contain benign lesions, **e–h** contain fibroadenomas, **i–l** contain ductal carcinoma in situ (DCIS), and **m–p** contain invasive lesions. The first column (**a, e, i, m**) contains the standard mammogram presentation, the second (**b, f, j, n**), third (**c, g, k, o**), and fourth (**d, h, l, p**) columns contain the corresponding three-compartment breast (3CB) LWP thickness map. Colorbars adjacent to each 3CB map indicate thickness in centimeters where red indicates areas of high thickness and thickness decreases towards the color violet. Thickness ranges are normalized across each column. Yellow lines are radiologist delineations of where biopsies were taken from which lesion pathology was determined.

lines. This difference is also quantified and presented in Table 4. Malignant lesions have a lower lipid, higher water, and higher protein signature when compared to non-malignant types. Compositional difference between malignant and non-malignant lesions amplifies when moving further out into the surrounding tissue or towards outer region 3. All compositional differences between malignant and non-malignant lesions result in significant $p$ values, see Table 4.

The orange dot-dashed line, in Fig. 6, indicates triple-negative lesions, and the blue dashed line represents all receptor-positive lesions: estrogen receptor, progesterone receptor, human epidermal growth factor receptor 2, or any combination of the three. The lipid content of triple-negative cancers is less than the lipid content of hormone receptor-positive cancers for the second and third outer regions as indicated by the space between the orange and blue lines in Fig. 6. Water and protein content of triple-negative cancers also appear to be distinctly different than

hormone-positive cancers and that difference increases the further away we get from the lesion (i.e. the difference when evaluating the surrounding tissue 2 mm away from the lesion is smaller than when evaluating the region 6 mm away from the lesion). This indicates a gradient difference in the lipid and water content of the surrounding tissue for triple-negative cancers. The change in the vertical position of the orange and blue lines across the three graph columns further demonstrates this gradient and suggest that receptor-positive cancers have different compositional gradients than triple-negative cancers.

## Discussion
This work suggests further support to the hypothesis that different breast lesion pathologies result in unique LWP compositions that can be directly measured through our 3CB, dual-energy mammography technique. We developed a neural network model

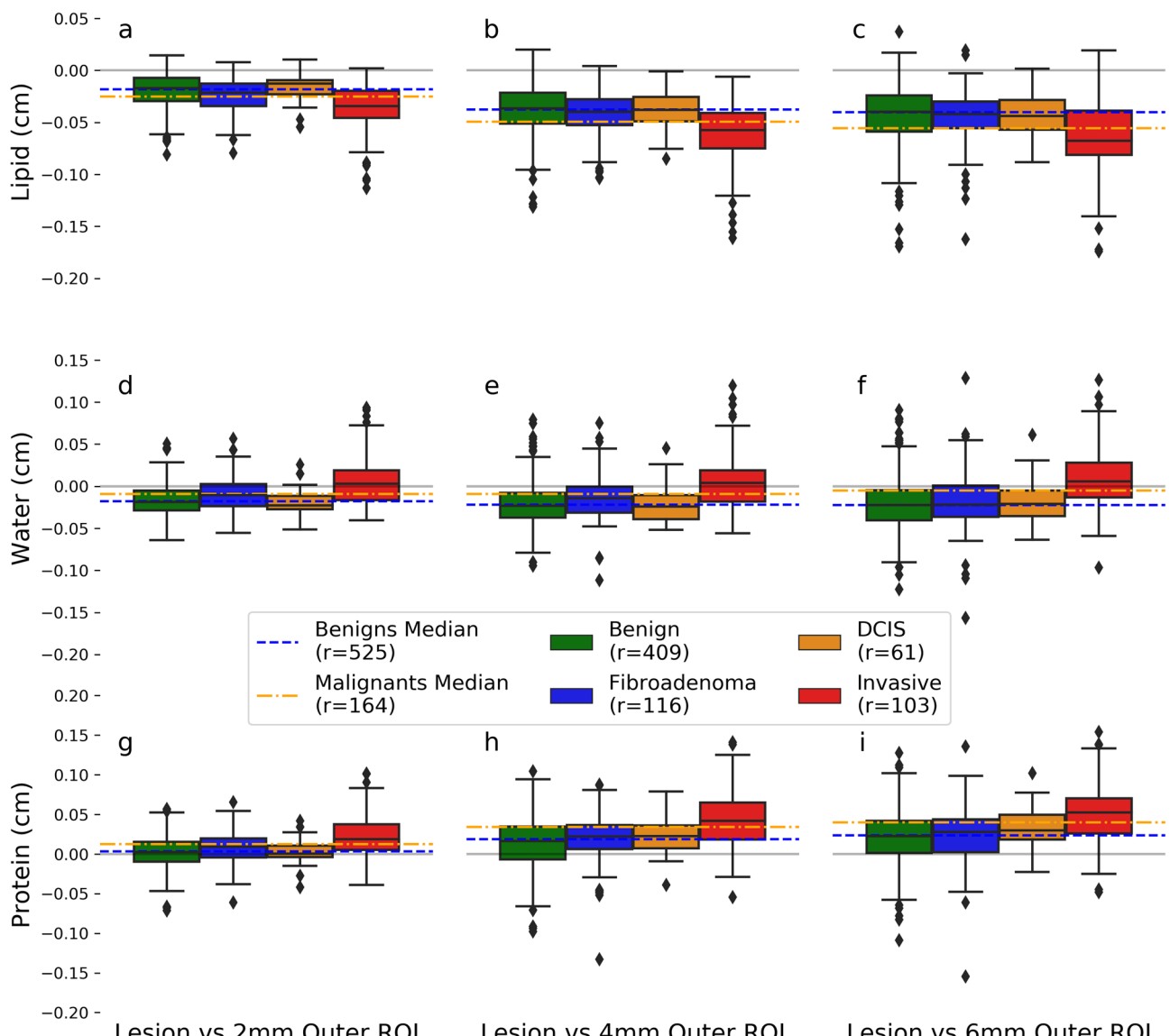

**Fig. 5 Lesion composition characterization.** Differences between median composition values of the lesion and the outer regions were calculated for all radiologist region of interest (ROI)s. Column one (**a**, **d**, **g**), two (**b**, **e**, **h**), and three (**c**, **f**, **i**) compares the composition within the lesion to the region 2, 4, and 6 mm from its border respectively. Each row looks at a different compositional component, lipid (**a**, **b**, **c**), water (**d**, **e**, **f**), or protein (**g**, **h**, **i**). Calculated differences are stratified by the following lesion types: benign (green), fibroadenoma (blue), ductal carcinoma in situ (DCIS) (orange), and invasive ductal carcinoma (invasive) (red). Median values of all benign lesions types, benign and fibroadenoma, and malignant types, DCIS and invasive, are represented by blue and orange dashed lines respectively. Gray line indicates zero and ROIs that lay above this line have more of a given composition when compared to its corresponding background outer region. Boxes represent the 25–75% interquartile range and the center line represents the median. Whiskers represents 1.5 the interquartile range and outliers which fall outside that range are depicted as diamonds.

**Table 4 Comparison between benign and malignant lesions.**

| Composition | Outer region | Malignant median | Benign median | Median difference | P value |
|---|---|---|---|---|---|
| Lipid | 1 | $-2.50e^{-02}$ | $-1.82e^{-02}$ | $-6.82e^{-03}$ | $1.37e^{-06}$ |
| Lipid | 2 | $-4.93e^{-02}$ | $-3.74e^{-02}$ | $-1.18e^{-02}$ | $7.49e^{-08}$ |
| Lipid | 3 | $-5.56e^{-02}$ | $-4.03e^{-02}$ | $-1.53e^{-02}$ | $8.61e^{-07}$ |
| Water | 1 | $-9.39e^{-03}$ | $-1.77e^{-02}$ | $8.36e^{-03}$ | $6.56e^{-07}$ |
| Water | 2 | $-9.17e^{-03}$ | $-2.21e^{-02}$ | $1.29e^{-02}$ | $2.43e^{-07}$ |
| Water | 3 | $-5.17e^{-03}$ | $-2.22e^{-02}$ | $1.70e^{-02}$ | $4.17e^{-08}$ |
| Protein | 1 | $1.23e^{-02}$ | $3.44e^{-03}$ | $8.87e^{-03}$ | $1.73e^{-08}$ |
| Protein | 2 | $3.42e^{-02}$ | $1.89e^{-02}$ | $1.52e^{-02}$ | $3.66e^{-09}$ |
| Protein | 3 | $3.99e^{-02}$ | $2.33e^{-02}$ | $1.66e^{-02}$ | $7.68e^{-10}$ |

Difference in compositions indicated by the space between blue and orange dashed lines in Fig. 5 are quantified in this table. P values were calculated using a Welch's test for unequal variance and all p values are significant, indicating that benign and malignant lesions have uniquely different compositions as measured by 3CB.

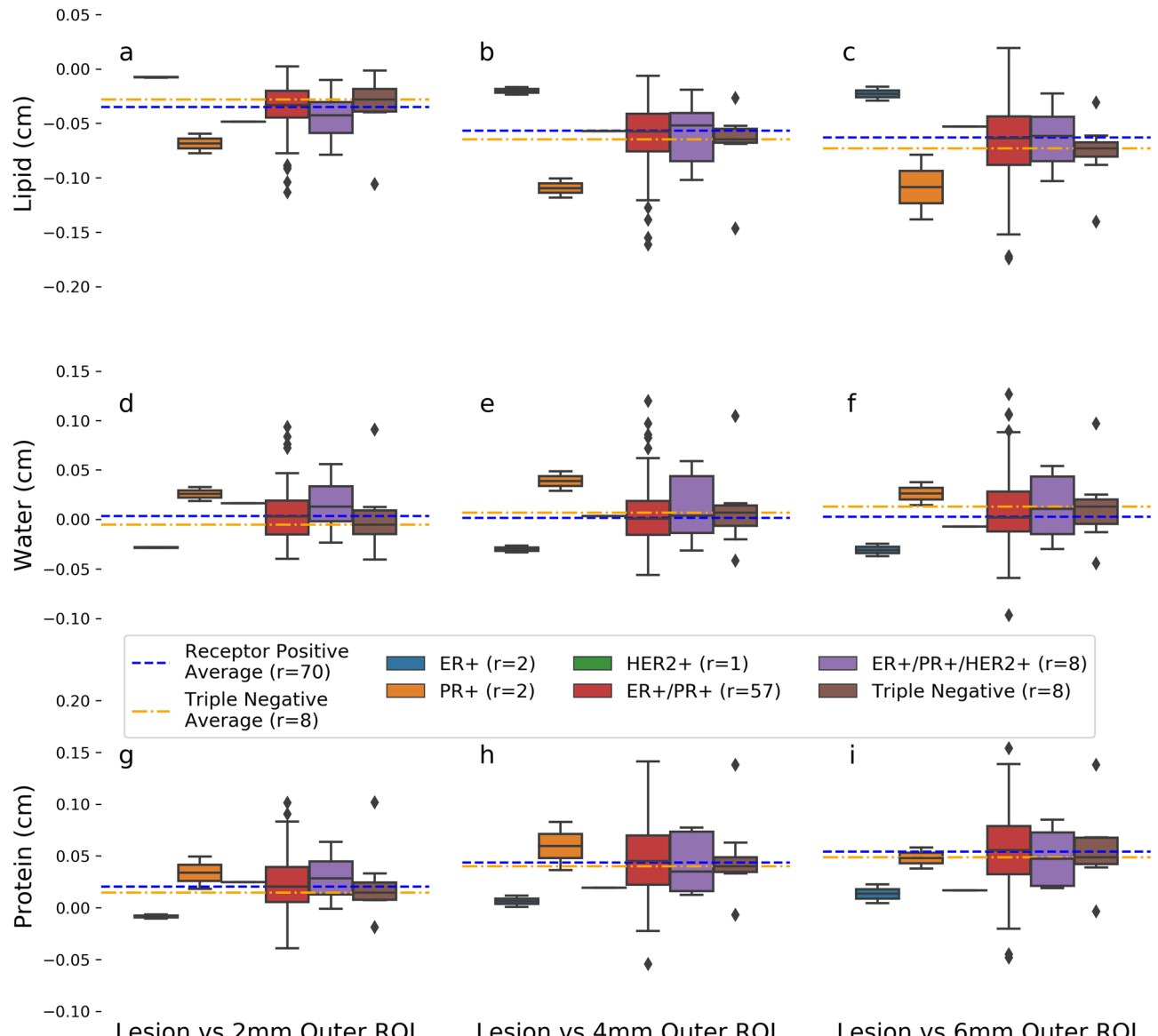

**Fig. 6 Hormone receptor composition characterization.** Compositional difference between invasive regions of interest (ROI) and their background are stratified by hormone receptor status. Each column of panels corresponds to the compositional signature at 2 (**a**, **d**, **g**), 4 (**b**, **e**, **h**), and 6 (**c**, **f**, **i**) mm from the lesion border while each row of panels corresponds to lipid (**a**, **b**, **c**), water (**d**, **e**, **f**), or protein (**g**, **h**, **i**). Median compositional difference between receptor-positive and triple-negative cancers are indicated by the blue and orange dashed lines, respectively. Receptor-positive median was calculated by including all lesion that contained either estrogen receptor (ER), progesterone (PR), or human epidermal growth factor receptor 2 (HER2) receptor status. Each subplot panel contains box and whisker plots for ER+ (blue), PR+ (orange), HER2+ (green), ER+/PR+ (red), ER+/PR+/HER2+ (purple), and triple-negative (brown) receptor statuses from left to right. Boxes represent the 25–75% interquartile range and the center line represents the median. Whiskers represents 1.5 the interquartile range and outliers which fall outside that range are depicted as diamonds.

to empirically demonstrate that adding compositional information improves classification of malignant and non-malignant lesions, providing diagnostic value. Further investigation into 3CB-derived features revealed distinct differences between lesions and their surrounding parenchyma composition, mechanisms that likely contribute to the increased predictive performance.

Previous work showed that compositional features were predictive of lesion pathology and models could be built to reasonably identify malignancies from composition alone[40]. Like trained radiologists, CAD software only has morphology, texture, and image opacity available to make malignancy probability decisions. When combining these morphologic features with compositional features from 3CB in our neural network model, the AUC on the test set increased. The IDI and NRI analysis showed that the

boost in AUC is attributed to increased specificity via the reduction of false positives or lowering malignancy probability on non-malignant lesions that CAD had previously assigned a high probability. The potential reduction in false positives is highlighted by the large red area between the new and reference models in Fig. 3b. Table 3 showed that the CAD + 3CB model was able to reclassify two ROIs to a BI-RADS category 3, which implies a potential to avoid unnecessary biopsies. This study specifically recruited BI-RADS 4 and 5 women and while we are not powered to extrapolate a strong conclusion about 3CBs ability to reclassify benign lesions to BI-RADS 3 or lower, our results suggest it to be a likely possibility. The addition of 3CB features also resulted in more accurate BI-RADS classification of malignant lesions and reclassification to lower BI-RADS categories for

non-malignant lesions, thus demonstrating an increase in confidence levels with respect to the decision to biopsy. Using 3CB imaging to increase specificity has the potential to be clinically beneficial with only minimal additional risk (10% additional dose from the acquisition of a second, high-energy mammogram). It should also be noted that this method of obtaining useful compositional information, unlike CEM, does not require contrast agents and the possibility of adverse reactions to contrast is non-existent.

Comparing compositional differences between lesions and their background supports the hypothesis different lesion pathologies present unique 3CB signatures. The aggressive biological nature of invasive lesions causes them to consume lipids at a high rate, and this phenomenon was observed on a macroscope scale with 3CB imaging. IDC's lower levels of lipid compared to surrounding tissue, observed in Fig. 5, is consistent with the literature. The lipid signature, which is the difference between the lesion and its surrounding region, increases as the region of comparison is moved further away from the lesion border. This further supports the aggressive growth natures of invasive cancers in that it begins to metabolize lipid from its peripheries. While non-malignant lesions were significantly different from malignant lesions for all composition types (LWP), it is likely that predictions were primarily driven by the lipid compositions since that signature is the greatest. There is a positive correlation between the magnitude of each compositional signature and the distance away from the lesion border. In other words, there is a gradient difference in tissue composition such that the composition becomes more different than normal breast tissue nearer the lesion. Although our models and analysis are focused on detection, gradient compositional changes of the breast could be useful in a screening situation as well. For instance, 3CB can be used to generate compositional gradient profiles of regions deemed suspicious on clinical for presentation FFDM. If the compositional profiles are similar to what we have observed in our study for malignant lesions, decreasing lipid and increasing water and protein, then the patient may be identified as high risk and subject to follow-up examination. Changes in compositional profiles and gradients within the breast are difficult to visualize on standard mammographic images alone due to the limited dynamic range of a single channel grayscale. 3CB images allow for independent assessments of possible changes in gradients for each tissue type, which is not possible on regular mammography. 3CB deconvolves the gradient into composition-specific gradients for lipid, water, and protein.

Recall, 108 3CB features were purposely extracted from the image in order to abstract the compositional information away from morphology. However, the entire 3CB thickness maps of LWP affords more information, on many orders of magnitude, than what was captured by the 108 features we used. In addition, there are more powerful computer vision methods such as convolutional neural networks which could potentially open the door to better automated detection and screening with 3CB images.

A reader study demonstrated CAD's ability to improve radiologists' ability to detect breast cancers[41]. Since we demonstrated improvements to CAD prediction with 3CB, it is reasonable to presume that the addition of 3CB would also further improve radiologists' ability to accurately detect and classify lesions. Ideally 3CB, like CAD, would be integrated as a tool into the clinical workflow. This would allow radiologist to interrogate the composition of any region of the breast to both lower false positives identified by CAD On the other hand, 3CB could identify a region not flagged by CAD that is expressing a compositional signature similar to that of a malignancy. Nonetheless, the translational clinical benefit of 3CB of this study is the increased confidence in the decision to biopsy which has the potential to reduce unnecessary biopsies.

## Data availability

Source data for the main figures in the manuscript can be found in "Supplementary Data 1". Imaging data linked to extensive meta data could be used to identify a participant and are therefore not publicly available. The data that support the findings of this study are available from the corresponding author upon reasonable request through a data sharing agreement.

## Code availability

Custom 3CB software (version 2020) for image analysis and lesion delineation, and trained model, 3CB-CNN, are available at https://github.com/shepherd-lab/3cb_software_and_model/tree/V0.1.1, https://doi.org/10.5281/zenodo.461542142.

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

## Acknowledgements
The authors would like to thank the patients who participated in this study. The authors would also like to thank iCAD Inc. for providing the CAD server, software, and support. Lastly, the authors would like to acknowledge the National Cancer Institute No. R01 CA166945 grant and California Breast Cancer Research Program No. 18IB0042 for funding this study.

## Author contributions
Developed the concepts and designed the study: J.A.S., K.K., M.L.G., and H.I.G.; scans interpretation and delineated biopsy sites: H.I.G., B.N.J., and B.N.; 3CB software development: S.M. and J.A.S.; 3CB biomarker research and data analysis: L.L.L. and K.D.; data storage, upkeep, and curation: L.L.L. and T.W.; machine learning modeling and statistical analysis: L.L.L. and P.S.; manuscript drafting or manuscript revision for important intellectual content: all authors; approval of final version of submitted manuscript: all authors; agrees to ensure any questions related to the work are appropriately resolved: all authors.

## Competing interests
K.D. Activities related to the present article: disclosed no relevant relationships. Activities not related to the present article: receives royalties from Hologic. Other relationships: disclosed no relevant relationships. M.G. Activities related to the present article: disclosed no relevant relationships. Activities related to the present article: is a stockholder in R2/Hologic; is a co-founder in Quantitative Insights (now advisor to Qlarity Imaging); receives royalties from Hologic, GE Medical Systems, MEDIAN Technologies, Riverain Medical, Mitsubishi, and Toshiba; receives royalties through institution is a licensee on patents. Other relationships: disclosed no relevant relationships. J.S. Activities related to the present article: in kind equipment support from Hologic and iCAD. Activities not related to the present article: investigator-initiated grant from Hologic. Other relationships: disclosed no relevant relationships. All other authors have no competing interests to declare.

## Additional information



