## [Peer Review File · Communications Medicine]

Reviewers' comments:

Reviewer #1 (Remarks to the Author):

This study looks at the change in accuracy of classification of a sample of benign and malignant lesions on mammography which is achieved by adding compositional information to commercially available CAD (iCAD) outputs. This is done via an algorithm derived from a neural network model. The sample was made up of women with lesions on mammography classified as BIRADS 4 and 5 and scheduled for breast biopsy. They underwent 3 compartment breast imaging (3CB) which involves an extra mammographic view with a technique specialised to allow measurement of lipid, water and protein within a region of interest (ROI). Radiologists identified a ROI corresponding to the lesion scheduled for biopsy. 3CB maps of the ROI and surrounding zones of tissue were generated and the data combined with CAD marks overlapping those areas to generate train, validation and test sets. An algorithm was trained to generate measures of probability of malignancy and thus BIRADS classifications.

The main findings were that the accuracy of prediction of malignancy improved, accuracy moving the AUC on a ROC from mean 0.69 for CAD alone to mean 0.81 with combined data. There was reduction in the BIRADS classification for benign lesions.

Compositional data for lesions was also studied and showed increased protein within lesions compared to surrounding tissue and increased water in invasive cancers. There were also differences in the composition of cancers related to hormone and biomarker status.

The findings will be of interest to workers in the field of CAD and AI for breast imaging. At present they have not been tested as part of a clinical pathway. The authors have previously published work on the combination of radiomics and 3CB and shown the potential to improve PPV of biopsy.

Specific comments:

Introduction, Page 2 line 26, distortion is also an important mammographic sign.

Page 3 line 40 ref 18. the reference refers to MRI, please supply a reference relating to enhancement on CEM.

Page 3 line 43 missing word at the end of the line (? imaging?)

page 3 line 45 ? missing word after atomic

Methods, There is no separate methods section before the results, the methods are admixed with results and this is confusing. Please lay out methods first. There is a separate section headed methods after the discussion which seems to have more detail about aspects of the methods than in the earlier section but some of the methods descriptions are duplicated between these 2 sections. There should be a single clear methods section. If this is too long some of the detail could be moved to an appendix.

Results, Page 4 line 71, please confirm informed participant consent.

Also some information on patient selection regarding whether symptomatic or screening and whether consecutive or selected. and if so on what basis.

page 7, line 129. and Fig 1a there are a number of cancers missed by CAD, (18 IDC and 1 DCIS). These have not been included in the ROIs used for analysis, including them would presumably change the apparent accuracy of the algorithm, this should be addressed in results and in the discussion.

Fig 1a, There are also a considerable number of false positive CAD marks from areas away from the

lesions/ radiologist ROIs. How would these be dealt with in a clinical pathway and what affect would these have on the accuracy of the algorithm? Again suggest deal with this is discussion as well as being explicit about the number of false marks in the results.

Page 8 using the term non-event for benign lesions, these would be better called benign.

Page 8 line 160, do you mean red instead of read?

page 8 line 163 and Fig 3 b, I find this paragraph and the figure lack clarity, I think they show a trend to reduce BIRADS classification for lesions after analysis for benign lesions. Please confirm whether any lesions moved to BIRADS 3 from 4 or 5 as this would potentially mean they could avoid biopsy. Please also clarify how the new BIRADS classification was derived, was it just on the basis of the score for probability of malignancy?

Discussion

The issues of effect of false negative CAD marks and false positive CAD marks on the accuracy of the algorithm should be discussed. There should also be some discussion of how this algorithm might be used in a real clinical pathway.

Page 12 line 225, the authors state that gradient compositional changes could be useful in screening. It would be helpful if they could explain how? It would also be interesting to have some idea of whether the authors think the gradient they have observed correlates to any visible change in conventional imaging.

page 12 line 261 the authors conclude that their technique could improve CADs ability to increase radiologist's detection of breast cancers. The performance of CAD in screening is more nuanced as is mentioned in the introduction by the authors. This should be addressed in the discussion as the false positive and false negative marks have not been included in the analysis here.

Reviewer #2 (Remarks to the Author):

Recommendation: Accept (Major Revision)

Comments:

The paper explored three compartment breast (3CB) to improve malignancy detection. Experimental results are given for 349 patients between breast cancer (698 images) consisted of 103 IDC, 61 DCIS, 116 FA, and 409 BN. The result shows CAD software with 3CB improved malignancy predictions yielded AUC from 0.69 (only CAD) to 0.81, and a feasible avenue to have breast imaging biomarkers. The paper also identified that invasive breast cancers have a unique compositional signature characterized. Overall, the paper is well written and the topic is properly motivated in the introduction section. The provided references are exhaustive for the understanding of the matter presented.

Additional Questions:

1. How many experienced radiologists did the manual annotation of the ROI? Was ROI the contour of lesion?
2. Was the manual annotation of the ROI done on one slice (and if one which slice) from the whole

volume or all slices of the volume? 2D or 3D mammogram?

3. Please describe more detail about the input of the neural network model. What was the input of only CAD? How to add 3CB features into CAD or the neural network model? I would suggest (but not limited) to show the processing flow of neural network model.

4. Were CAD and CAD + 3CB use the same neural network model?

5. Why did the authors extract 3CB of nine measurements (mean, median, standard deviation, minimum, maximum, kurtosis, skew, total and percentage value of all pixels) instead of the whole ROI image as the input to the neural network model?

6. For the prediction results of the neural network model (CAD and CAD + 3CB) which accuracy, sensitivity and specificity were reported? Did you use cross validation?

7. For bootstrapping of AUC how many samples (or %) were selected randomly?

8. In figure 3b, why did you take the BI-RADS 3/4a, 4a/4b, 4b/4c, 4c/5 borders at risk threshold of 2%, 10%, 50%, and 95%? Why not another?

9. Please change yellow lines (circle) to another color in figure 4. I think that would be more clear.

10. In figure 4, what was the unit of color bar? Why different content (lipid, water and protein) have different maximum number as the unit of color bar?

We thank the reviewers for their insightful comments. Below are our point by point responses.

Reviewer #1:

1. Introduction, Page 2 line 26, distortion is also an important mammographic sign.
Thank for your suggestion. Added distortions to the list.
2. Page 3 line 40 ref 18. the reference refers to MRI, please supply a reference relating to enhancement on CEM. *We have added reference 19 and 20 in addition to reference 18 that discusses this issue of benign lesion enhancement and increasing false positive findings.*
3. Page 3 line 43 missing word at the end of the line (? imaging?) *We have completed the sentence with "...initial screening exam."*
4. page 3 line 45 ? missing word after atomic. *The reviewer is correct. It should read, "...atomic composition..". Text has been updated accordingly.*
5. Methods, There is no separate methods section before the results, the methods are admixed with results and this is confusing. Please lay out methods first. There is a separate section headed methods after the discussion which seems to have more detail about aspects of the methods than in the earlier section but some of the methods descriptions are duplicated between these 2 sections. There should be a single clear methods section. If this is too long some of the detail could be moved to an appendix. *We had initially followed the formatting suggested by Nature Medicine since it seems that Communications in Medicine doesn't have an established format. To respond to your suggestion, we have reformatted the paper to have the conventional introduction, methods, results, discussion sections, but left the more detailed descriptions of the machine learning modeling in a supplementary methods section.*
6. Results, Page 4 line 71, please confirm informed participant consent.
Thank you for pointing out this oversight. All participants were sequentially recruited from women participating in screening mammography and had a BI-RADS 4 or higher diagnostic classification. All women were consented to participate. The first paragraph of the methods section has been updated to reflect these changes.
7. Also some information on patient selection regarding whether symptomatic or screening and whether consecutive or selected. and if so on what basis. *Women were selected from both a screening and diagnostic mammography population. Those who were BI-RADS 4 or 5, based on imaging, were ultimately recruited.*
8. page 7, line 129. and Fig 1a there are a number of cancers missed by CAD, (18 IDC and 1 DCIS). These have not been included in the ROIs used for analysis, including them would presumably change the apparent accuracy of the algorithm, this should be addressed in results and in the discussion. *Thank you for the comment. ROIs missed by CAD were not excluded from the analysis but these ROIs were held exclusively to the training set. It is true that including these missed ROIs in the testing set, for which we report results on, would impact accuracy.*
9. Fig 1a, There are also a considerable number of false positive CAD marks from areas away from the lesions/ radiologist ROIs. How would these be dealt with in a clinical pathway and what affect would these have on the accuracy of the algorithm? Again suggest deal with this is discussion as well as being explicit about the number of false marks in the results. *In the methods, we mention that the CAD was set to the most sensitive setting, higher than that used in a clinical setting. Without 3CB the higher sensitivity of the CAD would be a nuisance to the radiologist, causing them to review more potential lesions and slowing down workflow. With 3CB, the added CAD sensitivity*

provides more potential candidates for the model to review and improve both the sensitivity and specificity of CAD alone. Since there were quite a few malignancies missed with the commercial CAD used in our study, this implies that the sensitivity could have been even higher. See our discussion section. Therefore, although CAD may identify a lesion as suspicious, its compositional signature may not be that of a malignant lesion pathology such that both a false positive biopsy and/or added review time of the radiologist.

10. Page 8 using the term non-event for benign lesions, these would be better called benign. *We updated the term non-event to benign throughout the paper.*
11. Page 8 line 160, do you mean red instead of read? *“Read line” should be “red line” and has been corrected.*
12. page 8 line 163 and Fig 3 b, I find this paragraph and the figure lack clarity, I think they show a trend to reduce BIRADS classification for lesions after analysis for benign lesions. Please confirm whether any lesions moved to BIRADS 3 from 4 or 5 as this would potentially mean they could avoid biopsy. Please also clarify how the new BIRADS classification was derived, was it just on the basis of the score for probability of malignancy? *Thank you for the feedback. You are correct. The IDI curve shows an improved specificity and that 3CB composition can lower the probability of malignancy for benign lesions. In our test set, a small number of benign lesions were assigned a new malignancy probability within the thresholds of BI-RADS 3. These few lesions would avoid biopsies however, we mention in our discussion the caveat that is associated with this situation. The study was designed to recruit BI-RADS 4 and 5 women. BI-RADS 3 women are not included since they do not have biopsy-proven diagnoses. With these strong findings, we can now target larger populations of BI-RADS 4a women to characterize the probability of transition of 4a to 3.*
13. The issues of effect of false negative CAD marks and false positive CAD marks on the accuracy of the algorithm should be discussed. There should also be some discussion of how this algorithm might be used in a real clinical pathway. *CAD is currently used to bring a suspicious characteristic in the breast to the attention of the radiologist. It does this by using an internal model for probability of malignancy. However, the radiologist has to make their own decision on the probability and whether to biopsy or not. In effect, the CAD probability is highly discounted by the radiologist relative to their own training. A CAD false negative, or missed cancers, is arguably more deleterious since the region is not highlighted to the radiologist. Overly high sensitivity though would slow down the workflow. Our 3CB modification to CAD would optimize the probability of malignancy to be more accurate, remove false positive CAD delineations, and thus produce probabilities less discounted by the radiologist. This would be accomplished using the additional information of composition not available to the radiologist without disrupting workflow or read times.*
14. Page 12 line 225, the authors state that gradient compositional changes could be useful in screening. It would be helpful if they could explain how? It would also be interesting to have some idea of whether the authors think the gradient they have observed correlates to any visible change in conventional imaging.
Gradient grey-scale changes are currently used by detection algorithms to detect breast cancer. However, the dynamic range of a single channel image, such as a single energy mammogram in grey-scale, is more limited when compared to a three color channel image produced by 3CB. 3CB deconvolves the gradient into composition specific gradients for lipid, water, and protein. For example, a decreased lipid and increased

water and protein towards a particular region is indicative of malignancy. One can visualise the 3CB characteristics for malignancies to screen for potential malignancies. However, we have not tested to see if the image can be visually scored by a reader for accurate detection. This is a good idea and we can perform studies like this in the future as an extension of this work.

15. page 12 line 261 the authors conclude that their technique could improve CADs ability to increase radiologist's detection of breast cancers. The performance of CAD in screening is more nuanced as is mentioned in the introduction by the authors. This should be addressed in the discussion as the false positive and false negative marks have not been included in the analysis here. *Although this work focuses on detection improvements with 3CB, the results suggest possible utility in a screening scenario. When used in conjunction with CAD, 3CB can be used to profile the composition of the entire imaged breast. Further investigation into the composition of CAD identified regions may impact the false positive rate and specificity. On the other hand, 3CB could identify a region not flagged by CAD that is expressing a compositional signature similar to that of a malignancy and this could impact the screening false negative rates. In this study, CAD false negatives and CAD false positives that overlapped with radiologist ROIs were included in this analysis but held exclusively to the training set.*

Reviewer #2:

16. How many experienced radiologists did the manual annotation of the ROI? Was ROI the contour of lesion? *There were four radiologists that did the delineations. Yes, the entire contour of the lesion was delineated.*
17. Was the manual annotation of the ROI done on one slice (and if one which slice) from the whole volume or all slices of the volume? 2D or 3D mammogram? *All acquired images were 2D full field digital mammograms. ROIs were delineated on both CC and MLO views.*
18. Please describe more detail about the input of the neural network model. What was the input of only CAD? How to add 3CB features into CAD or the neural network model? I would suggest (but not limited) to show the processing flow of neural network model. *FFDM were the only input to the CAD and the CAD output ROIs each with its own probability of malignancy. The 108 extracted 3CB features and the CAD probability of malignancy was input into the neural network model. We have added a supplemental figure which details the CAD and neural network inputs and outputs.*
19. Were CAD and CAD + 3CB use the same neural network model? *They are different models. The CAD models utilizes a proprietary algorithm developed by iCAD Inc. while the CAD+3CB model is a neural network model.*
20. Why did the authors extract 3CB of nine measurements (mean, median, standard deviation, minimum, maximum, kurtosis, skew, total and percentage value of all pixels) instead of the whole ROI image as the input to the neural network model? *3CB compositional features were extracted from ROIs as an attempt to abstract composition away from morphology. An aim of this work was to show how composition adds useful information to existing tools and/or models which utilize morphological information. Whole ROI images contain shape and size information which we did not want the neural network to base its decision on.*
21. For the prediction results of the neural network model (CAD and CAD + 3CB) which accuracy, sensitivity and specificity were reported? Did you use cross validation?

We report on the area under the receiver operating characteristic (ROC) curve (AUC), the integrated discrimination improvement and the net reclassification index. Cross validation was not used and instead a train, validation, and test split was performed stratified by lesion type as to ensure that the proportions of malignancies and benigns were equivalent throughout each of the three datasets. Test set was held out and not seen by the model to avoid overfitting. The data split also ensured that ROIs from the same patient did not end up in different datasets and cause data leakage.

22. For bootstrapping of AUC how many samples (or %) were selected randomly?
The number of bootstrap samples as well as the resampling was performed randomly. Therefore the number of randomly selected samples each round differed and was only constrained by a minimum of one randomly selected resample.
23. In figure 3b, why did you take the BI-RADS 3/4a, 4a/4b, 4b/4c, 4c/5 borders at risk threshold of 2%, 10%, 50%, and 95%? Why not another? *These probabilities are directly associated with the American College of Radiology's (ACR) 5th edition BI-RADS lexicon. We are illustrating our models predictive performance in the context of clinical guidelines and recommendations.*
24. Please change yellow lines (circle) to another color in figure 4. I think that would be more clear. *Yellow lines delineations have been changed to black.*
25. In figure 4, what was the unit of color bar? Why different content (lipid, water and protein) have different maximum number as the unit of color bar?
The color bars are in units of centimeters and the range is standardized by tissue type. Each tissue type has different maximum numbers because each tissue type has a different range of thickness. I.E. None of the ROIs had a protein thickness above 1 cm while lipid thicknesses are seen in a range up to 7cm. If we kept the tissue types in the same range we would lose the dynamic color range of one or more tissue types. If we kept the range, say, at a maximum of 7 cm, we would barely see any difference in protein thickness and the image would be primarily indigo and violet.

REVIEWERS' COMMENTS:

Reviewer #1 (Remarks to the Author):

Thank you for the revisions. generally this now reads well.

I have a few further minor comments.

Page 7 line 177 you have repeated IP and IP, I think one of these should be IS

page 10 line 248 Typo: Error reference

Page 17 line 311 ? subspinous ? typo

Reviewer #2 (Remarks to the Author):

Comments:

No further comments. The refinements added by the authors partially improve the paper

We thank the reviewer and editor for their insightful comments. Below are our point by point responses.

Editorial Request:

1. Please see the attached document for editorial requests for the final version (.docx file). *We have responded to all request and suggested edits. We updated the request document with the details of our edits.*

Reviewer #1:

1. Page 7 line 177 you have repeated IP and IP, I think one of these should be IS. *Thank you for finding this and it has been corrected.*
2. Page 10 line 248 Typo: Error reference *The error was a broken link and is now corrected.*
3. Page 17 line 311 ? subspinous ? typo. *The word was corrected to be "suspicious".*

Reviewer #2:

No further comments. The refinements added by the authors partially improve the paper.